# Targeting prostate cancer by new bispecific monocyte engager directed to prostate-specific membrane antigen

Gargi Das[1,2], Jakub Ptacek[1], Jana Campbell[1], Xintang Li[3], Barbora Havlinova[1], Satish kumar Noonepalle[3], Alejandro Villagra[3], Cyril Barinka[1], Zora Novakova[1]*

1 Laboratory of Structural Biology, Institute of Biotechnology of the Czech Academy of Sciences, Vestec, Czech Republic, 2 Department of Cell Biology, Faculty of Science, Charles University, Prague, Czech Republic, 3 Lombardi Comprehensive Cancer Center, Georgetown University, Washington, D.C., United States of America

* zora.novakova@ibt.cas.cz

## Abstract

Prostate cancer (PCa) ranks as the second leading cause of cancer-related deaths among men in the United States. Prostate-specific membrane antigen (PSMA) represents a well-established biomarker of PCa, and its levels correlate positively with the disease progression, culminating at the stage of metastatic castration-resistant prostate cancer. Due to its tissue-specific expression and cell surface localization, PSMA shows superior potential for precise imaging and therapy of PCa. Antibody-based immunotherapy targeting PSMA offers the promise of selectively engaging the host immune system with minimal off-target effects. Here we report on the design, expression, purification, and characterization of a bispecific engager, termed 5D3-CP33, that efficiently recruits macrophages to the vicinity of PSMA-positive cancer cells mediating PCa death. The engager was engineered by fusing the anti-PSMA 5D3 antibody fragment to a cyclic peptide 33 (CP33), selectively binding the Fc gamma receptor I (FcγRI/CD64) on the surface of phagocytes. Functional parts of the 5D3-CP33 engager revealed a nanomolar affinity for PSMA and FcγRI/CD64 with dissociation constants of $K_D = 3\,nM$ and $K_D = 140\,nM$, respectively. At a concentration as low as 0.3 nM, the engager was found to trigger the production of reactive oxygen species by U937 monocytic cells in the presence of PSMA-positive cells. Moreover, flow cytometry analysis demonstrated antibody-dependent cell-mediated phagocytosis of PSMA-positive cancer cells by U937 monocytes when exposed to 0.15 nM 5D3-CP33. Our findings illustrate that 5D3-CP33 effectively and specifically activates monocytes upon PSMA-positive target engagement, resulting in the elimination of tumor cells. The 5D3-CP33 engager can thus serve as a promising lead for developing new immunotherapy tools for the efficient treatment of PCa.

## Introduction

Prostate cancer (PCa) remains one of leading causes of death amongst men. According to cancer statistics in 2023, PCa alone accounts for 29% of newly diagnosed cases and 12% of cancer-related deaths [1]. Hence, PCa management is one of the pressing unmet medical needs.

**Data availability statement:** All relevant data are within the article and its Supporting information files.

**Funding:** This work was supported by the Czech Academy of Sciences (RVO: 86652036), the Grant Agency of Charles University (GA UK project Number 358321 awarded G.D.), the Ministry of Education, Youth and Sports (LUAUS23254 awarded C.B.), and National Institutes of Health (NIH project R01CA249248 awarded A.V.). We acknowledge Imaging Methods Core Facility at BIOCEV, institution supported by the MEYS CR (LM2023050 Czech-BioImaging) and BIOCEV Biophysical Techniques CF of CIISB, Instruct-CZ Centre, supported by MEYS CR (LM2023042). The funders had no role in study design, data collection and analysis, decision to publish, or preparation of the manuscript. There was no additional external funding received for this study.

**Competing interests:** The authors have declared that no competing interests exist.

**Abbreviations:** ADA, anti-drug antibody; ADCP, antibody-dependent cell-mediated phagocytosis; BiTE, bispecific T-cell engager; CAR T, chimeric antigen T-cell receptor; CDC, cell-dependent cytotoxicity; Fc, crystallizable fragment of antibody; FcγRI, crystallizable fragment gamma receptor I; FDA, US Food and Drug Administration; GCPII, glutamate carboxypeptidase II; LPS, lipopolysaccharide; mAb, monoclonal antibody; mCRPC, metastatic castration-resistant prostate cancer; M-CSF, macrophage colony-stimulating factor; PCa, prostate cancer; PSA, prostate specific antigen; PSMA, prostate specific membrane antigen; ROS, reactive oxygen species; scFv, single chain variable fragment.

Prostate-specific membrane antigen (PSMA, EC 3.4.17.21), also known as glutamate carboxypeptidase II (GCPII), is an established PCa biomarker. PSMA is a 100 kDa type-II transmembrane protein with a large extracellular part that can be readily targeted by both small-molecule ligands as well as macromolecules and nanoparticles [2–6]. While PSMA is at low levels present in several healthy tissues including brain, kidney, salivary glands, prostate, and small intestine, PSMA expression levels are markedly increased in all stages of PCa, with the highest expression observed in metastatic androgen-resistant PCa [7–14]. Altogether, highly specific and abundant expression makes PSMA an ideal target for PCa imaging and therapy [15–19].

Owing to the success of immunotherapy targeting various hematological malignancies, immunotherapy approaches are now researched and utilized as treatment strategies against solid tumors, including prostate cancer [20–23]. For example, Sipuleucel-T, a cell-based vaccine exploiting patients' autologous dendritic cells loaded with prostatic acid phosphatase, was approved by the FDA in 2010 for the treatment of metastatic castrate-resistant hormone-refractory PCa [24,25]. Furthermore, several viral-based vaccines, including PROSTVAC, CV301, and Ad5-PSA, that trigger an immune response directed at PCa antigens are under investigation [26–29]. Yet another therapeutic approach involves blocking inhibitory signals for cytotoxic T-cells, as exemplified by Ipilimumab, which attenuates CTLA-4 function, thus enhancing the PCa antitumor effect of T-cells [30,31].

Monoclonal antibodies (mAb) targeting PSMA also attract current interest in the PCa field since they might offer superior specificity against PSMA compared to small molecule inhibitors [32]. mAbs can directly block proliferation of tumor cells [33] when applied as carriers of cytotoxic payloads. Moreover, they can elicit the host immune system to execute antitumor activity via cell-dependent cytotoxicity (CDC) or antibody-dependent cell-mediated phagocytosis (ADCP) [34–37]. The antibody-dependent activation of the complement pathway is an additional mechanism for eradicating abnormal tumor cells [38–43]. Similarly to PSMA-targeted immunotherapeutic strategies, PSMA-radioligand therapy using PSMA-specific antibody loaded with Lutetium-177 (177Lu-J591) was associated with a highly significant decrease in prostate-specific antigen (PSA) levels in metastatic castration-resistant prostate cancer (mCRPC) patients [44]. Furthermore, other mAb-based therapies including CAR T (chimeric antigen T-cell receptor) and BiTE (bispecific T-cell engager) targeting PSMA, are being intensively studied [45–49].

In this study, we report on the development and functional characterization of a bispecific monocyte engager capable of simultaneously targeting PSMA-positive cancer cells and the Fc gamma receptor I (FcγRI/CD64) receptor present on the monocyte/macrophage surface. The PSMA-targeting arm, in the form of the single-chain variable fragment (scFv), has been derived from the 5D3 antibody that revealed picomolar affinity and high specificity for human PSMA [49,50]. The CP33 cyclic peptide reported previously [51] was included to form fusion partner that targets FcγRI/CD64 receptors on monocytes/macrophages. In our hands, the 5D3-CP33 engager activated monocytes in a PSMA-dependent manner at concentrations as low as 150 pM leading to the killing of PSMA-positive cancer cells. The data suggest that the 5D3-CP33 fusion protein can serve as a promising candidate for developing future immunotherapeutic modalities targeting PCa.

## Results

### Expression and purification of 5D3/CP33 fusion proteins

5D3/CP33 monocyte engagers were designed using sequences of a single chain 5D3 fragment (scFv HL) and the CP33 sequence that were described previously [50,51]. To evaluate

the importance of the positioning of PSMA and CD64-targeting arms on expression levels, stability, and affinity of 5D3/CP33 fusions, we constructed two variants of monocyte engagers that differ in the arrangement of 5D3- and CP33-derived sequences. Both variants further harbor the BiP secretion signal and the SA-strep II affinity tag at their N- and C-terminus, respectively (Figs 1 and S1). Recombinant fusion proteins were heterologously expressed in S2 insect cells and purified from conditioned medium using the combination of Streptactin-XT affinity and size exclusion chromatography. A chimeric 5D3 molecule (ch5D3), comprising 5D3 variable murine domains fused to constant domains of human IgG1, was constructed to serve as a positive functional control (Fig 1C). Chimeric 5D3 was heterologously expressed in HEK-293T cells and purified via protein A affinity chromatography.

## Biophysical and functional characterization of 5D3/CP33 constructs

The purity and oligomeric status of purified fusions were evaluated by SDS-PAGE and the analytical size exclusion chromatography, respectively, confirming the presence of mono-disperse monomeric proteins with over >95% purity (Fig 1A-D). Next, the thermal stability of constructs was determined using differential scanning fluorimetry (nanoDSF) and a single peak corresponding to the melting temperature of 53.5 ˚C and 51.5 ˚C was observed for the 5D3-CP33 and CP33-5D3 construct, respectively (Fig 1E). nanoDSF analysis confirmed the existence of fully folded 5D3/CP33 fusions and obtained values are in line with melting temperatures reported previously for the 5D3-scFv HL construct [50].

Since the 5D3-CP33 fusion revealed slightly higher thermal stability (by 2 ˚C) than CP33-5D3 construct, the former was selected for ensuing functional experiments. Two peaks were observed for the ch5D3, where the first peak at 66.8 ˚C reflected the melting temperature of the Fab fragment, while the second peak at 82.2 ˚C corresponded to the third constant domain (CH3) of human immunoglobulin [52].

## Binding affinity of 5D3-CP33 and ch5D3 constructs

The binding specificity and affinity of each arm of the 5D3-CP33 fusion and ch5D3 were determined by flow cytometry using PSMA-positive PC3-PIP cells and FcγRI-positive HEK-293T-CD64 cells together with PC3, and HEK-293T cell lines included as PSMA- and FcγRI-negative controls, respectively. At 100 nM concentration, both 5D3-CP33 and ch5D3 bound selectively to PSMA- and CD64-positive cell lines, while nonsignificant binding was observed for corresponding controls, confirming the high selectivity of the constructs for both target antigens (Fig 2). Furthermore, 5-fold dilutions series of the constructs were used to determine binding affinities of individual arms for their respective antigens (Fig 2). For the 5D3 arm, dissociation constants of 3.4 nM and 1.6 nM were determined by the binding curve fitting for 5D3-CP33 and ch5D3 construct, respectively. Similarly, $K_D$ values of 140.4 and 2.2 nM were calculated for the anti-CD64 arms of 5D3-CP33 and ch5D3, respectively (Fig 2).

## Monocyte activation by 5D3-CP33 and ch5D3 constructs

The 5D3-CP33 fusion shall be able to activate monocytes/macrophages via binding to and clustering FcγRI/CD64 receptors on cell surface, and the activation status can be determined by quantifying the production of reactive oxygens species (ROS) by FcγRI/CD64-positive cells. U937 monocytic (pre-macrophage) cell line was implemented in cell-based experiments as the FcγRI/CD64-positive cell line that has been used in several studies as a phagocyting human model cell line [53,54], while prostate cancer-derived PC3-PIP and PC3 cells were used as representatives of PSMA-positive and PSMA-negative cells, respectively [55,56]. Prior to the experiment, U937 monocytes were stimulated by 0.1 μg/mL IFN-γ treatment running for 24

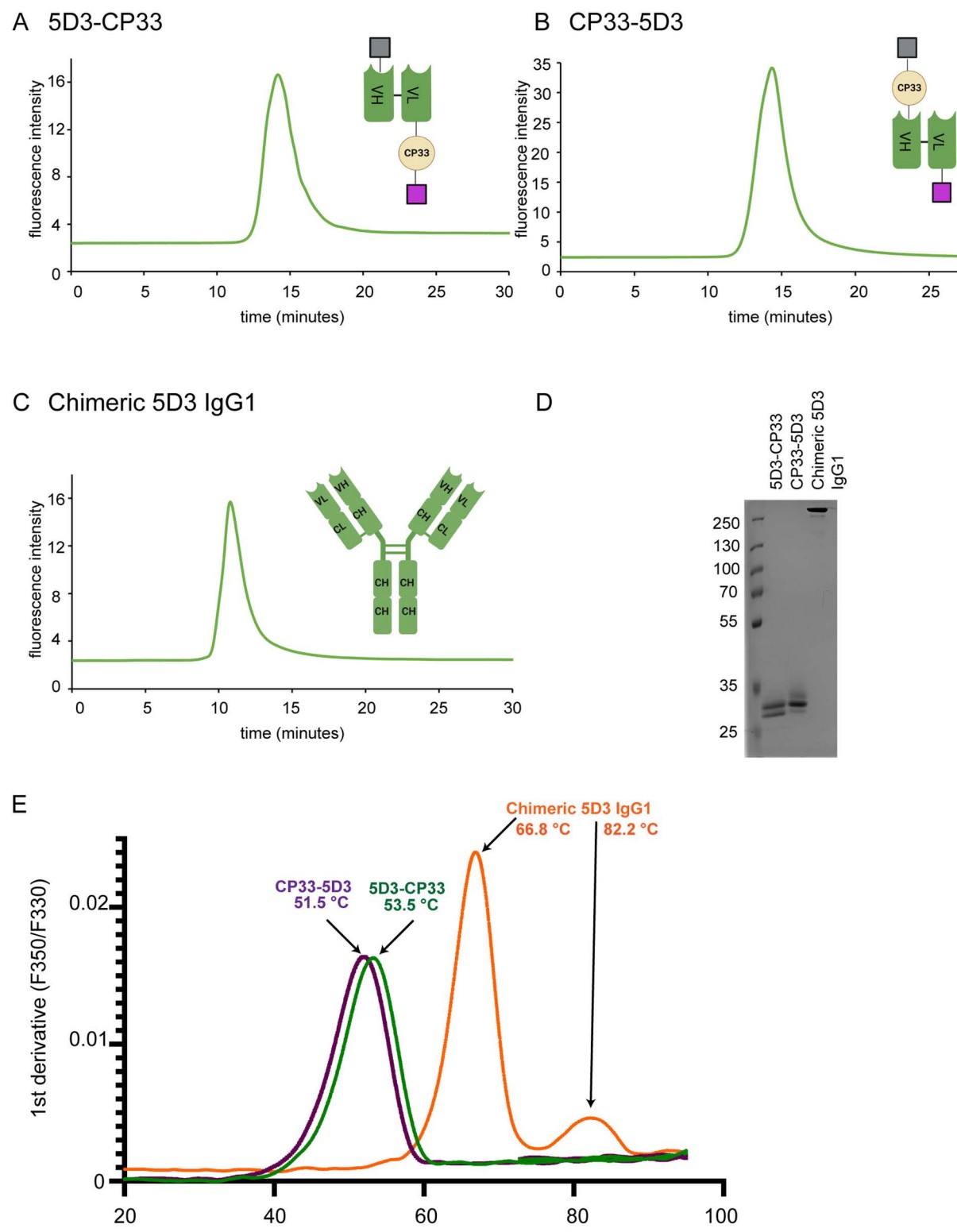

**Fig 1. Biophysical characterization of 5D3 monocyte engagers.** (A, B, C) A single peak in the chromatograms of analytical SEC confirmed a monomeric form of each construct. The schemes of constructs provided on the right side of each chart show arrangement of BiP signal (grey), antibody domains (green), CP33 (beige), and SA-strep II tag (violet; created with BioRender.com). (D) Non-reduced samples separated in SDS PAGE

gel stained by Coomassie Brilliant Blue G-250 show > 95% purity of all constructs. (E) The thermal stability of each construct was determined using nanoDSF. Melting temperatures of individual constructs were calculated from the first derivative of the fluorescence ratio at 350 nm and 330 nm (F350/F330).

hours to enhance FcγRI receptor levels on the cell surface. The stimulated U937 cells were then mixed with PC3-PIP or PC3 cells in the presence of 5-fold dilution series of the 5D3-CP33 construct (final concentration 200 nM to 0.32 nM), and ROS production was quantified by a lucigenin-based chemiluminescence readout. As shown in Fig 3, we observed a marked increase in ROS production by stimulated U937 cells upon mixing with PSMA-positive PC3-PIP cells in the presence of the 5D3-CP33 fusion, even at concentrations as low as 320 pM. At the same time, no detectable ROS production by activated U937 cells was observed in the presence of PSMA-negative PC3 cells or in the absence of the 5D3-CP33 fusion. These results thus confirm that U937 monocytes are selectively activated only upon simultaneous engagement of PSMA and CD64 antigens on the surface of cancer and immune cells, respectively. Similarly to 5D3-CP33, the ch5D3 construct selectively activated U937 cells only in the presence of PSMA-positive cells but not in the presence of PSMA-negative controls (Fig 3).

## Phagocytosis induced by 5D3-CP33 and ch5D3 constructs

In addition to ROS production, immune cells can eliminate target cancer cells by phagocytosis. To determine whether the 5D3-CP33 fusion can elicit selective phagocytosis of cancer cells managed by monocytes, IFN-γ-stimulated U937 and target PC3/PC3-PIP cancer cells were at first labeled with the DiD and DiO dye, respectively. U937 cells were then mixed with target cells and construct, and co-cultured for 1 hour. Following co-cultivation, phagocytosis was monitored by flow cytometry and confocal microscopy. Flow cytometry two-dimensional dot-plots (Fig 4A) were engaged to show specific signal arising from U937 cells, and PC3 or PC3-PIP cells, while double positive objects (upper right quadrant of the dot plot) were suggested to present process of engulfing target cells by U937 cells. Flow cytometry analysis revealed that antibody-dependent cell-mediated phagocytosis (ADCP) is only evident under conditions, where U937 cells are co-cultured with PSMA-positive PC3-PIP cells in the presence of 5D3-CP33. In the presence of 111 nM 5D3-CP33, approximately 25% of double-positive cells correspond to cancer cells being engulfed by stimulated monocytes. Conversely, a limited number (<2%) of double-positive cells is observed in the absence of 5D3-CP33 or when PSMA-negative PC3 cells were used. Fig 4B further shows a positive correlation between the level of ADCP and 5D3-CP33 concentrations ranging from 0.15 nM to 1 μM in U937/PC3-PIP co-cultures, revealing that 5D3-CP33 can induce cancer-cell phagocytosis at concentrations as low as 150 pM. Like 5D3-CP33, the ch5D3 construct demonstrated selective phagocytosis by U937 cells exclusively in the presence of PSMA-positive cells, with no activation observed in PSMA-negative controls (Fig 4C). To provide visual confirmation of flow cytometry data, we imaged ADCP in analyzed cell mixtures by confocal microscopy (S2 Fig). The microscopy data revealed co-localization of specific signals arising from U937 cells (red) and target cells (green) and confirmed engulfment of PC3-PIP cells by the activated monocytes, whereas no engulfed objects were observed in U937/PC3 mixtures.

To mimic physiological conditions in our phagocytosis assay more closely, we used human M1-polarized macrophages known to possess anti-tumor activity (as reviewed in [57–59]). Monocytes extracted from human peripheral blood mononuclear cells (PBMCs) were differentiated using a treatment with macrophage colony-stimulating factor (M-CSF) and then polarized by the addition of IFN-γ and lipopolysaccharide (LPS). The polarized macrophages were labeled with CellTrace CFSE (green color; Fig 4D) and

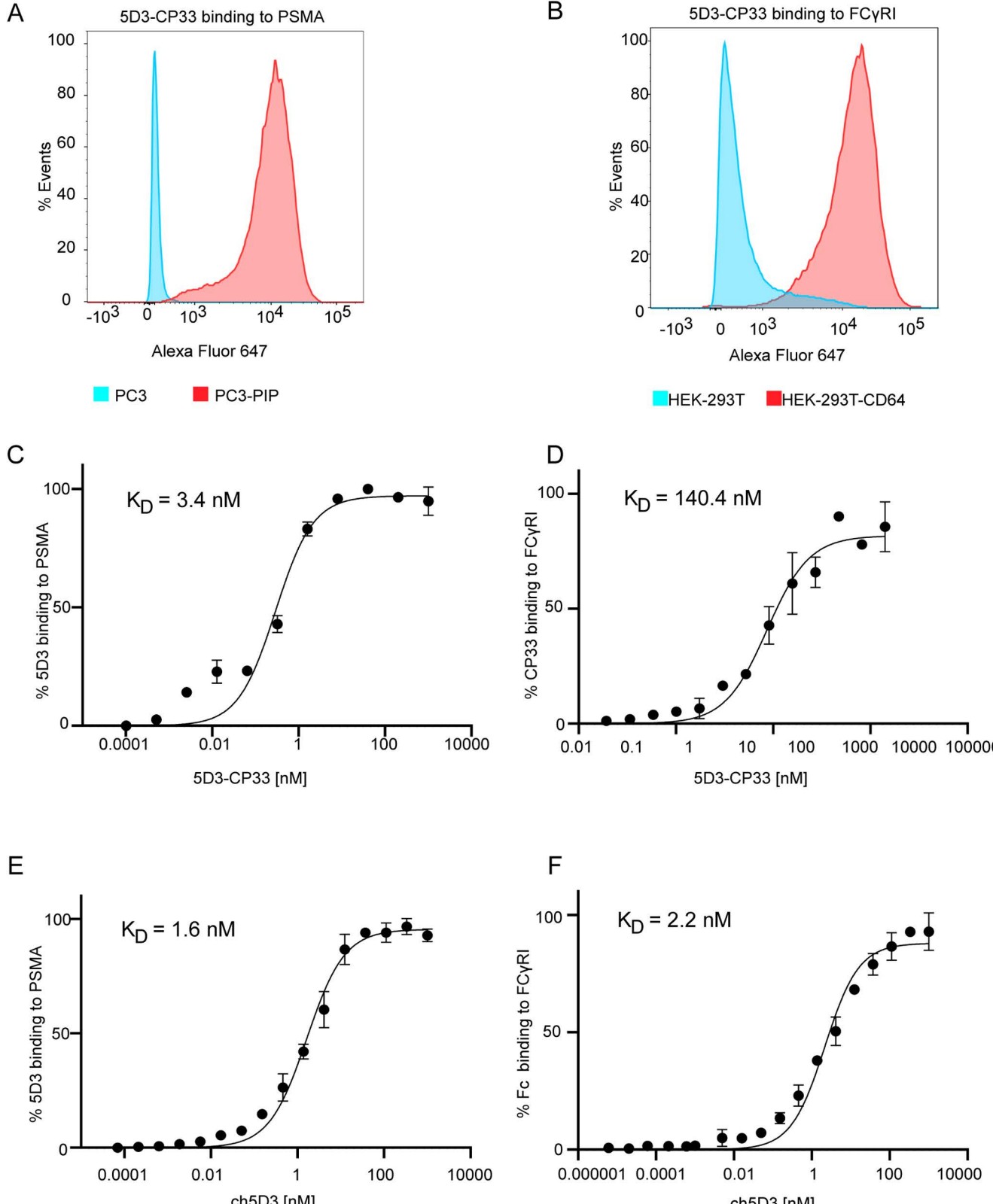

**Fig 2. Characterization of purified 5D3-CP33 and ch5D3.** (A) Target specificity of 200 nM 5D3-CP33 to PSMA antigen was determined by flow cytometry using PSMA-positive PC3-PIP cells. PC3 cells served as PSMA-negative control. Approximately 30,000 live cells were included in analysis to generate

histograms. (B) Specificity of 400 nM 5D3-CP33 to FcγRI/CD64 was estimated on CD64-positive HEK-293T-CD64 cells using flow cytometry. HEK-293T cells served as FcγRI/CD64-negative controls. Approximately 30,000 live cells were involved in the analysis. (C, D) Specific affinity of 5D3-CP33 to PSMA and FcγRI/CD64 was determined by flow cytometry using PC3-PIP cells and HEK-293T-CD64 cells, respectively. PC3 and HEK-293T cells were used as negative controls. (E, F) Estimation of specific affinity of ch5D3 run identically to 5D3-CP33 measurement. Individual $K_D$ values are shown in the upper left corner of charts.

combined with target cells pre-stained with CellTrace Far Red (red color) in the presence/absence of 5D3-CP33 (final concentration of 111 nM). Phagocytosis, represented by dually labeled objects (Fig 4D, white arrowheads), was observed in all samples, pointing towards a partial unspecific activation of M1 macrophages (Fig 4E). Importantly, the phagocytosis rate of PC3-PIP cells was significantly enhanced in the presence of 5D3-CP33, whereas no significant difference in phagocytosis rate was observed for PC3 in the presence/absence of 5D3-CP33. These data thus confirm PSMA-specific killing of tumor cells that is mediated by 5D3-CP33.

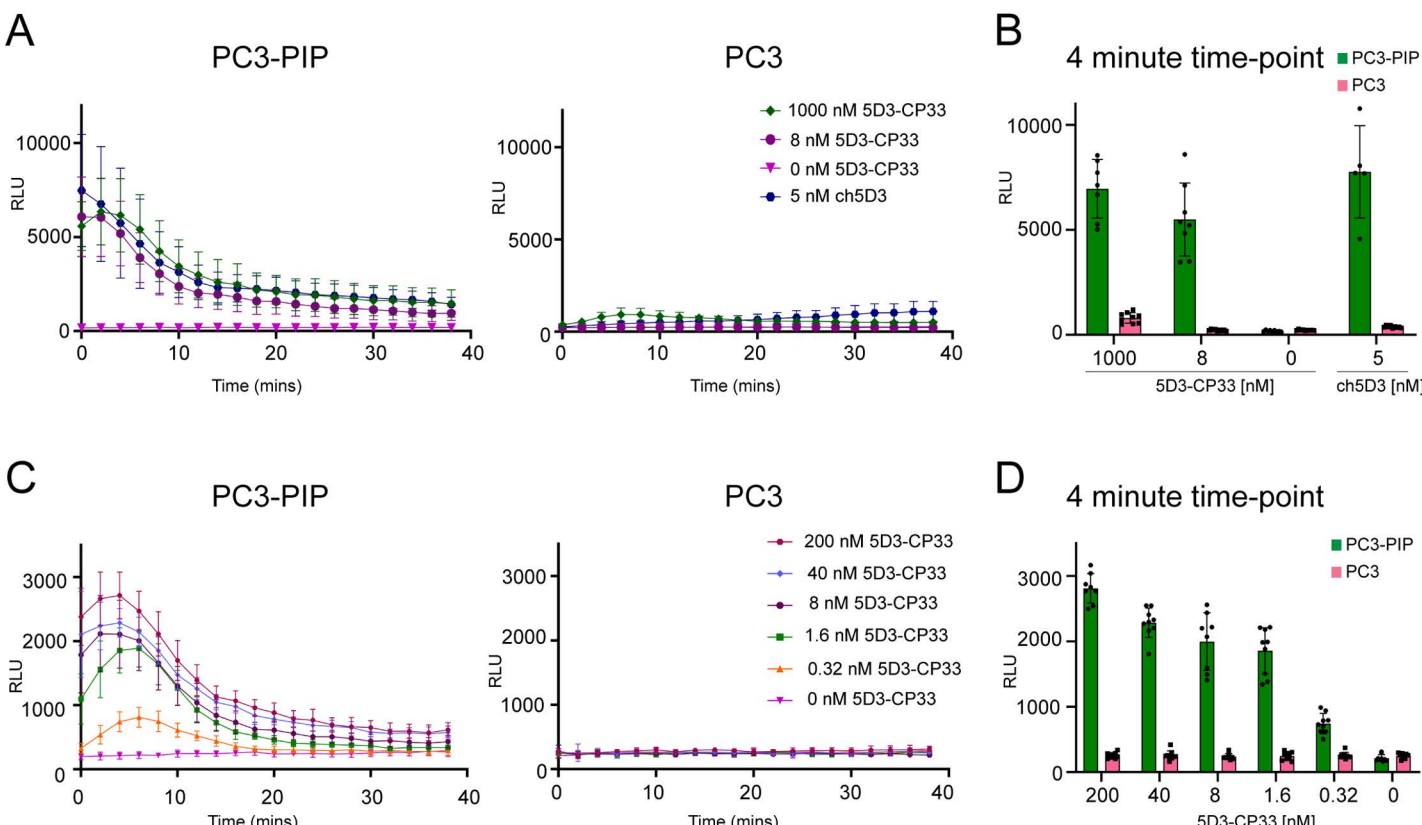

**Fig 3. ROS production driven by 5D3-CP33.** (A) Comparison of the effect of 5D3-CP33 and ch5D3 on ROS production by activated U937 cells co-cultured with PSMA-positive PC3-PIP cells or PSMA-negative PC3 cells. (B) Quantification of produced ROS was visualized at 4-minute time-point to show in detail the difference between various concentrations of constructs and the ratio of ROS production in the presence of PSMA-positive and PSMA-negative cells. (C) ROS production by activated U937 cells in the presence of serial dilution of 5D3-CP33 co-cultured with PC3-PIP cells or PC3 cells. (D) Visualization at 4-minute time-point shows in detail concentration-dependent effect of 5D3-CP33 on ROS production in the presence of PSMA-positive and negative cells. The level of ROS was measured using the lucigenin-based chemiluminescence assay.

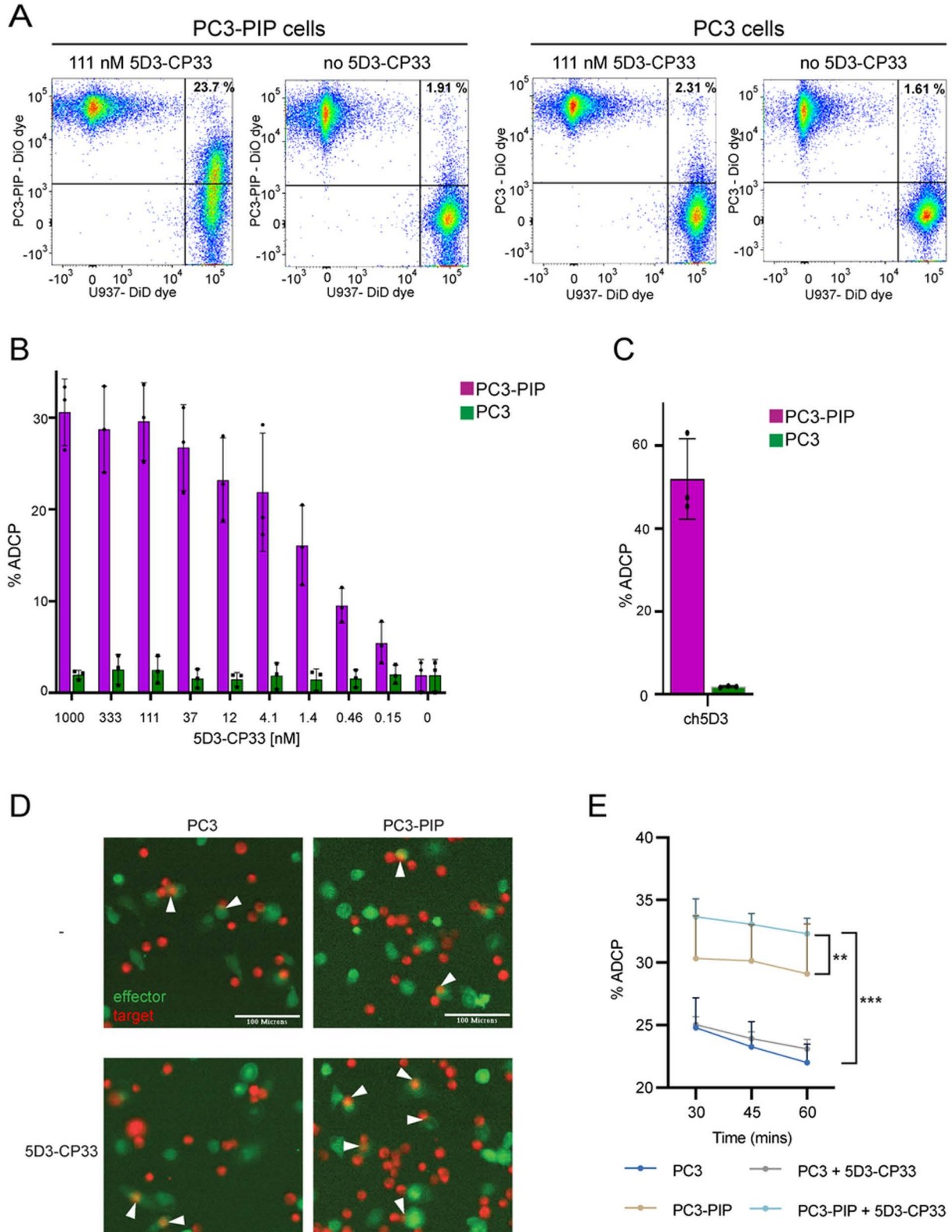

**Fig 4. Antibody-dependent cell-mediated phagocytosis (ADCP) of prostate cancer cells by U937 monocytes and human macrophages.**
(A) PC3/PC3-PIP cells and U937 cells were labeled by DiO and DiD dye, respectively, co-incubated in the presence/absence of 111 nM 5D3-CP33 and analyzed by flow cytometer. The upper right quadrant of two-dimensional dot-plot charts shows double-positive events

representing target cells engulfed by U937 monocytes. (B) U937/PC3-PIP co-cultures revealed a positive correlation between the level of ADCP and 5D3-CP33 concentration. (C) The chart shows a selective effect of 5 nM ch5D3 on the ADCP level in U937/PC3 and U937/PC3-PIP co-cultures, respectively. (D) Monocytes were isolated from human PBMCs and M1-polarized macrophages were derived by M-CSF and IFN-γ/LPS treatment. Macrophages (green) were mixed with PC3/PC3-PIP target cells (red) in the presence/absence of 111 nM 5D3-CP33. After 30 minute incubation samples were imaged by an automated digital fluorescence imaging system. Illustrative examples of phagocytosis are marked by white arrowheads. Scale bar: 100 µm. (E) Time-lapse chart shows quantification of phagocytosis events and statistics (parametric T-test; ** P value < 0.005, *** P value < 0.0005).

## Discussion

Myeloid cells, including monocytes and macrophages, can readily infiltrate the tumor microenvironment (TME), and they constitute a substantial portion (30-50%) of tumor-infiltrating immune cells [60,61]. Macrophages within the TME are referred to as tumor-associated macrophages (TAMs), and their phenotypes oscillate between M1 tumor suppressive and M2 pro-tumorigenic subtypes [62,63]. TAMs are prevalent in most cancers that reveal poor clinical outcomes [64,65]. The M2 macrophages promote cancer growth by supporting angiogenesis, metastasis, and directly interfering with effector T-cells at the tumor site [63,66]. A significant role of TAMs in tumor physiology provoked the development of therapeutic modalities focusing on these cells to improve clinical outcomes of cancer patients [60,67,68]. Several small molecules were designed to enhance macrophage anti-tumor activity. As an example, the R848 ligand was used to reprogram TAMs into the M1 tumor-suppressive phenotype thereby potentiating an effect of the ADCP treatment [69,70]. Additionally, small molecule inhibitors of the CCR2 receptor on macrophages could enhance chemotherapy effects in models of pancreatic ductal adenocarcinoma [71].

Inspired by the success of the chimeric antigen receptor T-cell (CAR-T) in leukemia treatment, CAR-macrophages engineered for the immunotherapy of solid tumors have garnered attention in recent years [72–75]. CAR-macrophages, advantageously derived from patients' inducible pluripotent stem cells (iPSCs) [76,77], can convert M2 macrophages to the M1 tumor-suppressive phenotype without reverse transformation [78]. These cells can support T-cell cytotoxicity by generating proinflammatory signals and upregulating MHC and TNF expression [79]. This study aimed to reveal an alternative macrophage-mediated immunotherapy that simultaneously engaged FcγRI/CD64 receptors and prostate-specific membrane antigen present on the surface of host macrophages and prostate cancer cells, respectively. The 5D3/CP33 bispecific macrophage engagers were designed to eradicate tumor cells via host immune cells without need of technically challenging and economically demanding engineering of personalized CAR-macrophages.

Direct comparison of the 5D3/CP33 engager with the full length chimeric 5D3 antibody could make the estimation of their performance *in vitro* more accurate. Both the CP33 peptide and the antibody crystallizable fragment (Fc) bind to overlapping epitopes on FcγR receptors, with the former being strictly selective for FcγRI over other FcγR variants [51]. At the same time, an apparent affinity of CP33 for FcγRI is approximately 63-fold lower than the affinity of Fc part of the intact antibody (140.4 *vs* 2.2 nM, respectively; our data and ref [51]). Despite its markedly lower, mid-nanomolar, FcγRI affinity, the 5D3/CP33 engager can still elicit macrophage activation (ROS production, phagocytosis) at picomolar concentrations *in vitro*. The ability of the 5D3/CP33 engager to mediate targeted elimination of PSMA-positive cancer cells at particularly low concentrations would be beneficial *in vivo*, where lower dosage can minimize adverse immunogenic effects such as the anti-drug antibody (ADA) response or the cytokine release syndrome [80,81]. Additionally, small molecular size of the 5D3/CP33 fusion (35 kDa) compared to the full-length antibody (150 kDa) can be advantageous due to better penetrability in the solid tumor micro-environment

[82,83]. On the other hand, the longer circulation time of the full-length antibody may be preferable for higher therapeutic efficacy [84–88]. It is obvious that optimal fine-tuned functional characteristics of the engager can only be determined in (pre)clinical settings and might require engineering of 5D3/CP33 variants with extended serum half-lives and increased stability *in vivo* [89–92].

In 2014, McEnaney and colleagues reported on a synthetic antibody mimic targeting prostate cancer (SyAM-P) representing a fully synthetic small-molecule fusion of CP33 [93]. PSMA-binding arm of the molecule was derived from a urea-based scaffold currently used in clinic for prostate cancer imaging and therapy [94]. Additional improvement of effector functions came with the construction of multivalent derivatives. In the case of SyAM-P as well as 5D3/CP33, affinity of the CP33 part for monocytes was found to be considerably low when compared to the affinity of PSMA-binding arm. However, the difference might carry benefits in limited binding to monocyte surface receptors in the absence of an antigen, minimizing thus potential off-target effects [95]. Interestingly, replacing the urea-based scaffold of SyAM-P by 5D3-scFv in the 5D3/CP33 engager has the potential to further mitigate off-target effects of the engager as the urea-based ligands were reported to engage other physiological targets, including glutamate carboxypeptidase III, NAALAdase L (PSMA paralogs) and the mGluR8 receptor [32,96].

## Conclusions

We developed a novel 5D3-CP33 engager with the potential to mediate targeted eradication of PSMA-positive prostate cancer cells by host immune system cells. Picomolar concentrations of the engager efficiently activate monocytes *in vitro* and elicit killing of target cancer cells by combining reactive oxygen species production and phagocytosis. Our data thus encourage further optimization and investigations including *in vivo* testing. Moreover, the engager is suggested to serve as a promising candidate for developing future immunotherapeutic modalities targeting prostate cancer and other solid tumors expressing the PSMA antigen.

## Materials and methods

### Chemicals and reagents

The chemicals and reagents were purchased from Sigma-Aldrich (Steinheim, Germany) unless stated otherwise. The restriction enzymes and ligases were purchased from New England Biolabs (Ipswich, MA, USA).

### Cell culture and cell lines

PC3-PIP and PC3 cell lines were kindly provided by Dr. Warren Heston (Cleveland Clinic, Cleveland, OH, USA) [9], and suspension culture of HEK-293T cells was kindly donated by Dr. Ondrej Vanek (Faculty of Science, Charles University, Prague, Czech Republic). U937 and adherent HEK-293T cell lines were acquired from the American Type Culture Collection (ATCC, Gaithersburg, MA, USA). PC3-PIP, PC3 and U937 cells were maintained in RPMI-1640 media, whereas adherent HEK-293T cell lines were cultivated in Dulbecco's Modified Eagle media. Both cultivation media were supplemented with 10% v/v fetal bovine serum (FBS; Gibco, Life Technologies, Carlsbad, CA) and 2 mM L-glutamine (Life Technologies, Thermo Fisher Scientific, Carlsbad, CA, USA), and cells were maintained at 37 °C under 5% $CO_2$ atmosphere. Suspension culture of HEK-293T cells was cultivated in the 1:1 mixture of FreeStyle F17 (Gibco) and EX-CELL 293 media. Insect Drosophila melanogaster Schneider S2

cells were cultivated at 26°C in Insect-XPRESS media (Lonza, Basel, Switzerland) supplemented with 2 mM L-glutamine.

## Construction of 5D3/CP33 fusions

Two variants of monocyte engagers were created in this study. In 5D3-CP33 variant, 5D3-scFv HL (heavy-light chain) was fused to the N-terminus of CP33 [51], while design of CP33-5D3 variant contained opposite order of functional parts. The 5D3-scFv-CP33 gene string and the opposite variant were amplified by PfuUltra II Hotstart PCR Master Mix (Agilent, Santa Clara, CA, USA) using 0.8 μM primers specified in S1 Table. The amplification program started by initial denaturation (95 °C for 2 min) followed by 25 cycles of denaturation (95 °C for 30 s), annealing (60 °C for 30 s) and extension (72 °C for 1 min) finalized by extension at 72 °C for 10 min. Amplified sequences were digested using BglII and XhoI enzymes, and the digested products were ligated into the backbone vector pMT/BiP/V5-HisA. Final plasmids obtained were termed pMT/BiP/5D3-CP33 and pMT/BiP/CP33-5D3, representing the 5D3-CP33 and CP33-5D3 fusions, respectively.

## Construction of chimeric 5D3

Variable domains of 5D3 mAb and backbone of expression vector pVITRO1-dV-IgG1/κ (a gift from Andrew Beavil; Addgene plasmid #52213) [97] were amplified by Phusion Flash High-Fidelity PCR Master Mix (Thermo Fisher Scientific) using primers specified in S1 Table. Amplification of 0.5 μg template started by initial denaturation (98 °C for 10 sec) followed by 25 cycles of denaturation (98 °C for 1 s), annealing (60 °C for 5 s) and extension (72 °C for 75 s) and finalized by extension at 72 °C for 1.5 min. PCR products purified by GenElute PCR Clean-Up Kit (Sigma-Aldrich) were mixed in molar ratio 1:1 and 0.37 pmol of total DNA was then treated by Gibson Assembly Master Mix (New England Biolabs) for 3 hours at 50°C. Assembled molecules were transformed into XL1-Blue competent cells (Agilent) and final expression plasmid was isolated using QIAGEN Plasmid Midi Kit (Qiagen, Hilden, Germany).

## Stable overexpressing cell lines

pMT/BiP/5D3-CP33 and pMT/BiP/CP33-5D3 expression vectors were transfected into S2 cells together with the selection plasmid pCoBLAST (Invitrogen) as described earlier [50]. Transfected cultures were selected using 40 μg/mL blasticidine (InvivoGen, San Diego, CA, USA) until stably transfected cell population was established.

Expression plasmid carrying gene of human FcγRI/CD64 receptor (pCDNA4-CD64) was kindly provided by Pavel Sacha (Institute of Organic Chemistry and Biochemistry, Prague, Czech Republic). The vector was introduced into adherent HEK-293T cells using JetPRIME transfection reagent (Polyplus, Illkirch, France). FcγRI/CD64-positive cells were selected in media containing 50 μg/mL Zeocin (InvivoGen). Single-cell colonies were isolated using cloning discs and further continuously cultivated in the presence of 50 μg/mL Zeocin.

## Expression and purification of constructs

Monocyte engagers were over-expressed by stably transfected S2 cells in 7 days upon induction by 0.7 mM $CuSO_4$. The conditioned media was then harvested, filtered by tangential flow filtration (Sartoflow Smart, Sartorius Stedim Systems GmbH, Guxhagen, Germany), and purified by streptactin-XT affinity purification according to the established protocol [50]. Further, eluted fractions from the affinity chromatography column were subjected to size exclusion chromatography (SEC) using an Enrich 70 10/300 column (GE Healthcare Biosciences, Uppsala, Sweden), connected to the NGC Chromatography System (Bio-Rad Laboratories,

Hercules, CA, USA). The mobile phase used in SEC was phosphate buffered saline (PBS) supplemented by 3% glycerol.

Expression vector of chimeric 5D3 was introduced in suspension culture of HEK-293T cells using linear polyethylene imine as described previously [98]. Five days after transfection, conditioned medium of the culture was harvested by sequential centrifugation at 500xg for 10 mins and 10 000xg for 30 mins, respectively. Supernatant was filtered by tangential flow filtration, mixed with preequilibrated protein A agarose (Pierce, Thermo Fisher Scientific) and incubated 1 hour at RT. Agarose was then stringently washed by PBS supplemented with 1 mM EDTA and 10% glycerol. Protein was eluted by 100 mM glycine pH 2.7 and immediately neutralized. Chimeric 5D3 was filtered, flash frozen at concentration 2 mg/mL in liquid nitrogen and stored at -80°C until further use.

### Thermal Stability by nanoDSF

Proteins at final concentration 0.3 mg/mL were subjected to a temperature gradient from 25 °C to 95 °C using a Prometheus NT.48 fluorimeter (NanoTemper Technologies, München, Germany). The melting temperature (Tm) was calculated from the first derivative of a fluorescence ratio emitted at 350 nm and 330 nm.

### Determination of binding affinities

Harvesting of cells and the detection of interaction between an engager and a specific antigen was performed as described in [50]. In short, cells were mixed with protein of interest in a total volume of 20 μL and incubated for 15 min at 4°C. Presence of 5D3-CP33 on the surface of cells was detected by sequential staining with the mouse monoclonal anti-Strep tag antibody (1 μg/mL, Immo, IBA; cat.no. 2-1517-001) and a goat anti-mouse secondary antibody conjugated to Alexa Fluor 647 (0.25 μg/mL; Thermo Fisher Scientific; cat.no. A-21236). Binding of 5D3-CP33 on HEK-293T-CD64 and HEK-293T cells was detected by Strep-TactinXT conjugated to DY-649 (0.1 μg/mL; Immo, IBA; cat.no. 2-1568-050). Bound ch5D3 was detected by goat anti-human secondary antibody conjugated to Alexa Fluor 647 (1 μg/mL; Thermo Fisher Scientific; cat.no. A-21445). All incubations were carried out 15 minutes at 4°C followed by stringent washes. The fluorescence signal was acquired by BD LSRFortessa flow cytometer (BD Biosciences, San Jose, CA, USA). Data were analyzed using the FlowJo software (FlowJo, LLC, Ashland, OR, USA). Dissociation constants ($K_D$) for each arm were calculated in the GraphPad Prism software using a non-linear regression algorithm (Graph-Pad, San Diego CA, USA).

### Quantification of ROS production

U937 monocytes were activated by 0.1 μg/mL IFN-γ overnight at 37 °C. Following incubation, $15 \times 10^4$ effectors were mixed with targets at the ratio 1:1 in a white 96-well U-bottom microplate (Nunc, Thermo Fischer, MA, USA), then 5D3-CP33 was added to cell suspension. Stock of lucigenin (1.2 mg/mL in RPMI-1640 media) was added to wells to reach final concentration 0.12 mg/mL, making a total volume of reactions 100 μL. The plate was centrifuged at 200xg 2 min and the chemiluminescence signal of lucigenin was measured in regular intervals (135 s) for 45 minutes in a Clariostar microplate reader (BMG Labtech, Ortenberg, Germany) pre-heated to 37 °C.

### Antibody-dependent cell-mediated phagocytosis

U937 effector monocytes were activated by 0.1 μg/mL IFN-γ overnight at 37 °C. The following day, monocytes and target cells were incubated with 1.9 μM Vybrant DiD Cell-Labelling solution and 5.7 μM DiO Cell Labelling solution (Thermo Fisher Scientific), respectively.

Incubation run in serum-free RPMI-1640 media at 37 °C for 30 minutes in cell density $1 \times 10^6$ cells/mL. Following incubation, cells were washed three times with RPMI-1640 media supplemented with 14% low IgG FBS (Gibco, Thermo Fisher Scientific). Target and effector cells were mixed in ratio 1:1 (total number $30 \times 10^4$ cells per well, total volume 75 uL) in 96-well U-bottom microplate. Cells were then mixed with 3-fold dilution series of 5D3-CP33 ranging from 1 μM to 0.15 nM in final volume 100 μL. The plate was centrifuged at 200xg for 2 min and incubated at 37 °C for 1 hour. Following incubation, the plate was cooled to 4°C and samples were analyzed with a BD LSR Fortessa flow cytometer.

In microscopy analysis, cell samples were processed by the same procedure described above. Prior imaging cells were dissolved in cooled FluoroBrite DMEM media (Thermo Fisher Scientific), applied onto glass coverslips precoated by poly-L-lysine and immediately analyzed by a Leica TCS SP8 confocal microscope (Leica, Wetzlar, Germany) equipped with a water immersion objective HC PL APO CS2 with magnification 63x. Microscope instrumentation was preequilibrated to 20°C. The set of images was acquired in z-axis at 2 μm intervals. Images were analyzed in Fiji [99] and processed in CS4 Photoshop software (Adobe Systems, San Jose, CA).

Culture of human macrophages was prepared by differentiation of monocytes obtained from human PBMCs using recombinant human M-CSF (BioLegend, San Diego, CA, USA) to stimulate monocytes. Briefly, monocytes were isolated using the EasySep™ Human Monocyte Isolation Kit (Stemcell Technologies, Vancouver, Canada) following the manufacturer's protocol. Monocytes were incubated with M-CSF (50 ng/mL) for 5 days in cultivation medium RPMI-1640 supplemented with 10% v/v FBS, 1% w/v non-essential amino acids, 1% w/v penicillin/streptomycin, and 5% v/v human serum (Our Blood Institute, Oklahoma City, OK, USA). After this period, cultivation medium was exchanged, and cells were kept in culture another 24 hours. Macrophages were then harvested by scraping from 10-cm cell culture dish and seeded into a black flat-bottom 96-well plate at the concentration 10 000 cells/well. Macrophages were cultivated another 24 hours to let them attach to the well bottom. Then they were polarized to the M1-like phenotype by IFNγ (50 ng/mL) and lipopolysaccharide (10 ng/mL) treatment running for 48 hours. Following polarization, the macrophages were stained in-plate with CellTrace CFSE Cell Proliferation Kit (Thermo Fisher Scientific). The tumor cells were harvested by trypsinization and stained with CellTrace Far Red Cell Proliferation Kit (Thermo Fisher Scientific) following the manufacturer's instructions. Stained tumor cells (13 000 cells/well) were co-cultured with macrophages in effector: target ratio 0.8:1 along with the addition of 111 nM 5D3-CP33 bispecific engager. Image acquisition was performed with Molecular Devices ImageXpress Pico Automated Cell Imaging System (Molecular Devices, San Jose, CA, USA). Data analysis was performed with Molecular Devices ImageXpress software (Version 2.9.3, Molecular Devices) using the phagocytosis module to compute the phagocytosis rate. Statistical analysis was performed using a parametric T-test in GraphPad Prism software (GraphPad).

## Supporting Information

**S1 Table. List of primers.**
(PDF)

**S1 Fig. Sequence of 5D3/CP33 constructs.**
(PDF)

**S2 Fig. Microscopy images of phagocytosis.**
(PDF)

**S1 Raw Images. Raw image of SDS PAGE gel.**
(PDF)

## Acknowledgement

We thank I. Jelinkova for her outstanding technical assistance.

## Author contributions

**Conceptualization:** Gargi Das, Jakub Ptacek, Cyril Barinka, Zora Novakova.

**Data curation:** Gargi Das.

**Formal analysis:** Gargi Das, Jana Campbell, Xintang Li, Satish kumar Noonepalle, Cyril Barinka.

**Funding acquisition:** Gargi Das, Alejandro Villagra, Cyril Barinka.

**Investigation:** Gargi Das, Jakub Ptacek, Jana Campbell, Xintang Li, Barbora Havlinova, Satish kumar Noonepalle, Zora Novakova.

**Methodology:** Gargi Das, Satish kumar Noonepalle.

**Project administration:** Cyril Barinka.

**Resources:** Cyril Barinka.

**Supervision:** Alejandro Villagra, Cyril Barinka, Zora Novakova.

**Writing – original draft:** Gargi Das, Jakub Ptacek, Xintang Li, Cyril Barinka, Zora Novakova.

**Writing – review & editing:** Alejandro Villagra.

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
