## [Decision Letter · Decision Letter 0]

25 Jul 2024

PONE-D-24-27288Targeting prostate cancer by new bispecific monocyte engager directed to prostate-specific membrane antigenPLOS ONE

Dear Dr. Novakova,

Thank you for submitting your manuscript to PLOS ONE. After careful consideration, we feel that it has merit but does not fully meet PLOS ONE’s publication criteria as it currently stands. Therefore, we invite you to submit a revised version of the manuscript that addresses the points raised during the review process.

We look forward to receiving your revised manuscript.

Kind regards,

Pier Carlo Ricci

Academic Editor

PLOS ONE

Journal Requirements:

"This work was supported by the Czech Academy of Sciences (RVO: 86652036), the Grant Agency of Charles University (GA UK project Number 358321 awarded G.D.), and the Ministry of Education, Youth and Sports (LUAUS23254 awarded C.B.). We acknowledge Imaging Methods Core Facility at BIOCEV, institution supported by the MEYS CR (LM2023050 Czech-BioImaging) and BIOCEV Biophysical Techniques CF of CIISB, Instruct-CZ Centre, supported by MEYS CR (LM2023042).

Additional Editor Comments:

Dear Dr. Zora Novakova,

We have now received the reviews from the peer reviewers.

One of the reviewers has provided feedback indicating that the paper is suitable for publication in its current form. However, the second reviewer has made some constructive comments and suggestions for improvement that we believe would enhance the quality and impact of your manuscript.

I recommend that you carefully consider and incorporate the revisions suggested by the second reviewer. These revisions are intended to strengthen your arguments, clarify certain points, and improve the overall presentation of your research.

Please find the detailed comments and suggestions of the second reviewer attached to this letter. We ask that you submit a revised version of your manuscript addressing these points, along with a response letter outlining the changes made and how you have addressed the reviewer's concerns. We believe that following the second reviewer's recommendations will significantly enhance the quality of your work

Thank you for your attention to this matter. We look forward to receiving your revised manuscript and are confident that the revisions will contribute to the excellence of your publication.

Kind Regards,

Carlo Ricci

Reviewers' comments:

Reviewer's Responses to Questions

**Comments to the Author**

1. Is the manuscript technically sound, and do the data support the conclusions?

Reviewer #1: Yes

Reviewer #2: Partly

2. Has the statistical analysis been performed appropriately and rigorously? 

Reviewer #1: Yes

Reviewer #2: N/A

3. Have the authors made all data underlying the findings in their manuscript fully available?

Reviewer #1: Yes

Reviewer #2: Yes

4. Is the manuscript presented in an intelligible fashion and written in standard English?

Reviewer #1: Yes

Reviewer #2: Yes

5. Review Comments to the Author

Reviewer #1: The paper aims to develop and functional characterize a bispecific monocyte engager capable of simultaneously targeting PSMA-positive cancer cells and the Fc gamma receptor I (FcγRI/CD64 ) receptor present on the monocyte/macrophage surface. As far as this paper is in my field of expertise (except technical aspects), the manuscript is well written, clear and of interest to readers, thanks to the increasing interest that PSMA has widely spread in recent years. Due to my limited experience in chemistry and radiochemistry, I cannot judge the technical parts.

Reviewer #2: The paper needs importante amendments

Demonstrate engagement of human macrophage

U937 alone Is not convincing

Perform functional experiments at different E/T ratios

A construct control lacking PSMA Binding is needed

6. PLOS authors have the option to publish the peer review history of their article (what does this mean? ). If published, this will include your full peer review and any attached files.

**Do you want your identity to be public for this peer review?** For information about this choice, including consent withdrawal, please see our Privacy Policy .

Reviewer #1: No

Reviewer #2: **Yes: ** Pierosandro Tagliaferri

---

## [Author Response · Author response to Decision Letter 1]

24 Jan 2025

Vestec, January 24, 2025

Editorial board of PLOS ONE

Dear editors,

Thank you very much for the decision letter concerning our manuscript entitled “Targeting prostate cancer by new bispecific monocyte engager directed to prostate-specific membrane antigen”. We appreciate assessment of our work by reviewers, and in the revised version of the manuscript, we addressed all major concerns.

According reviewers’ recommendations we added new experimental data related to human macrophages that further evaluate specificity end effect of 5D3-CP33 engager. We discussed all points given by reviewer 2 and proposed detailed explanation of controls used in study. We enhanced the information about effector:target ratio used in cell-based experiments. Changes to the manuscript and detailed explanation are provided in a point-by-point response to reviewers’ comments at the end of this letter. Corrections of grammar and stylistics are shown in track-changes version.

Authors kindly notify that the surname of one co-author Jana Nedvedova has been changed to „Campbell“ due to her wedding that happened during year 2024.

All gel image data are shown in the body of manuscript (Fig 1D), all original raw gel data are available in Supporting information file „S1_raw_images“ submitted together with revised manuscript.

Be so kind to update information in manuscript according amended Funding Statement: "This work was supported by the Czech Academy of Sciences (RVO: 86652036), the Grant Agency of Charles University (GA UK project Number 358321 awarded G.D.), the Ministry of Education, Youth and Sports (LUAUS23254 awarded C.B.), and National Institutes of Health (NIH project R01CA249248). We acknowledge Imaging Methods Core Facility at BIOCEV, institution supported by the MEYS CR (LM2023050 Czech-BioImaging) and BIOCEV Biophysical Techniques CF of CIISB, Instruct-CZ Centre, supported by MEYS CR (LM2023042). The funders had no role in study design, data collection and analysis, decision to publish, or preparation of the manuscript. There was no additional external funding received for this study."

We feel that we have provided a thorough response to reviewers’ and editor’s comments and, with their assistance, have strengthened the manuscript by addressing key concerns.

Many thanks for seeing our manuscript through the review process.

Sincerely,

Zora Novakova, PhD

Corresponding author

Laboratory of Structural Biology

Institute of Biotechnology of the Czech Academy of Sciences

Průmyslová 595

252 50 Vestec

Czech Republic

phone: +420-325 873 736

email: zora.novakova@ibt.cas.cz

Review Comments to the Author

Reviewer #1: The paper aims to develop and functional characterize a bispecific monocyte engager capable of simultaneously targeting PSMA-positive cancer cells and the Fc gamma receptor I (FcγRI/CD64 ) receptor present on the monocyte/macrophage surface. As far as this paper is in my field of expertise (except technical aspects), the manuscript is well written, clear and of interest to readers, thanks to the increasing interest that PSMA has widely spread in recent years. Due to my limited experience in chemistry and radiochemistry, I cannot judge the technical parts.

Reviewer #2: The paper needs importante amendments

Demonstrate engagement of human macrophage

U937 alone Is not convincing

Authors‘ answer: Based on reviewer’s suggestion, effect and specificity of 5D3-CP33 bispecific engager was evaluated on human macrophages differentiated from monocytes (isolated from human PBMCs). Data was inserted as panels D,E in Figure 4. The presence of 5D3-CP33 enhances specific phagocytosis of PSMA-positive but not PSMA-negative cells.

In this study, the U937 cell line was employed as a well-established and standardized model routinely used to evaluate phagocytosis. We believe that results related to isolated human macrophages that were added to revised manuscript strengthen main findings of the study.

Reviewer #2: Perform functional experiments at different E/T ratios

Authors‘ answer: We appreciate reviewer’s comment. Numerous previously published studies show specific phagocytosis of tumor cells driven by linkage via recombinant/synthetic molecules (see papers specified by DOI: 10.1002/ange.201510866, 10.1021/acschembio.0c00112, 10.1002/cpch.88, 10.1021/acscentsci.3c01052, 10.1016/j.biomaterials.2020.120601, 10.1080/19420862.2020.1857100, 10.4049/jimmunol.1402891, 10.1016/j.jim.2007.04.009, 10.1091/mbc.E03-09-0668, 10.18632/oncotarget.18492). In these studies, that utilized monocyte cell lines as well as freshly isolated human monocytes, the E/T ratio varied from 4:1 to 1:1 with significant preference of ratio 1:1. Since our data show superior rate of phagocytosis at E/T ratio 1:1 (Figure 4) that represents standard ratio used preferentially in experimental studies, we conclude that increase in the amount of effector cells would not bring any significant benefit to our study.

Specific phagocytosis-supporting effect of 5D3-CP33 was observable even when target cells were present in excess over effectors represented by freshly isolated human monocytes differentiated and polarized into M1 type macrophages (E/T ratio was approximately 0.8:1), see Figure 4E.

Reviewer #2: A construct control lacking PSMA Binding is needed

Authors‘ answer: We agree with reviewer that implementation of proper controls is critical for the study. The specific binding of both parts of the bispecific engager was evaluated in detail and verified on effector and target cells using flow cytometry (see data in Figure 2 of our study). To show specific effect of 5D3-CP33, we implemented in our study two types of negative control: 1) a mixture of cells with the absence of the bispecific molecule, and 2) PSMA-negative target cells highly similar to PSMA-positive target (PSMA-positive cells represent original cell line overexpressing recombinant PSMA). First negative control shows potential non-specific phagocytosis, whereas second negative control simulates the absence of PSMA binding and is equal to the construct lacking PSMA binding part as suggested by reviewer. The results clearly indicate that PSMA-negative cell line is not attracted by the macrophages, even though 5D3-CP33 engager is present in concentration as high as 1 µM in cell mixture. Both negative controls prove specific binding and stimulatory effect of 5D3-CP33 in phagocytosis of PSMA-positive cells. Additionally, our study includes IgG1 molecule with standard Fc part that serves as a positive control (see Figure 4 of our study), therefore readers could assess directly the efficiency of CP33 part of bispecific engager in binding and activation of macrophages.

---

## [Decision Letter · Decision Letter 1]

3 Feb 2025

Targeting prostate cancer by new bispecific monocyte engager directed to prostate-specific membrane antigen

PONE-D-24-27288R1

Dear Dr. Novakova,

We’re pleased to inform you that your manuscript has been judged scientifically suitable for publication and will be formally accepted for publication once it meets all outstanding technical requirements.

Kind regards,

Pier Carlo Ricci

Academic Editor

PLOS ONE

Additional Editor Comments (optional):

Reviewers' comments:

Reviewer's Responses to Questions

**Comments to the Author**

1. If the authors have adequately addressed your comments raised in a previous round of review and you feel that this manuscript is now acceptable for publication, you may indicate that here to bypass the “Comments to the Author” section, enter your conflict of interest statement in the “Confidential to Editor” section, and submit your "Accept" recommendation.

Reviewer #2: All comments have been addressed

2. Is the manuscript technically sound, and do the data support the conclusions?

Reviewer #2: Yes

3. Has the statistical analysis been performed appropriately and rigorously? 

Reviewer #2: N/A

4. Have the authors made all data underlying the findings in their manuscript fully available?

Reviewer #2: Yes

5. Is the manuscript presented in an intelligible fashion and written in standard English?

Reviewer #2: Yes

6. Review Comments to the Author

Reviewer #2: The authors correctly addressed all the points and the paper is now ready for pubblication. Data will add novel information in the field

7. PLOS authors have the option to publish the peer review history of their article (what does this mean? ). If published, this will include your full peer review and any attached files.

**Do you want your identity to be public for this peer review?** For information about this choice, including consent withdrawal, please see our Privacy Policy .

Reviewer #2: **Yes: ** Pierosandro Tagliaferri, MD

---

## [Editor Report · Acceptance letter]

PONE-D-24-27288R1

PLOS ONE

Dear Dr. Novakova,

I'm pleased to inform you that your manuscript has been deemed suitable for publication in PLOS ONE. Congratulations! Your manuscript is now being handed over to our production team.

Kind regards,

on behalf of

Prof. Pier Carlo Ricci

Academic Editor

PLOS ONE